# Resilience of Urban Technical Networks

**DOI:** 10.3390/e21090886

**Published:** 2019-09-12

**Authors:** Serban Raicu, Eugen Rosca, Dorinela Costescu

**Affiliations:** Polytechnic University of Bucharest, Splaiul Independentei 313, 060042 Bucharest, Romania

**Keywords:** urban system, network states dynamics, resilience, maximum entropy principle

## Abstract

The need to overcome the insulated treatment of urban technical infrastructures according to the nature of the transferred flows is argued. The operation of urban technical networks is affected by endogenous and exogenous random events with consequences for users. By identifying these operational risks and the difficulties of estimating the impact on the performance of the urban technical networks, the authors chose to study the risk management through a concise expression—in relation to the engineering resilience and its connections with vulnerability. Further, the research is confined to the case of urban traffic networks for which the resilience is expressed by the capabilities of these networks of resistance and risk absorption (both motivated by the redundancy in design and execution). The dynamics of the network, in correlation with the resistance and absorption capacities, is introduced by three states for which the signal graph is built. In a stationary regime, the probability of each state is computed. These probabilities allow the calculation of the entropy of the network, relevant for assessing the preservation of the network functionality.

## 1. Introduction

The city is a geotype of a societal substance based on coexistence: population, residences, activities, relationships, and networks [1,2,3]. The city, as a complex socio-technical dynamic system, consists of subsystems (technical infrastructures, residences, activities, and social relationships) that interact dynamically with each other, providing political, administrative, and economic functions [4]. 

Urban technical infrastructures, as well as those of the whole of society, consist of constructions (punctual elements), connected by a multitude of connections (linear elements) that ensure the relationships in which the transfer of material, energy, and informational flows are used for the benefit of the economic and social system [5].

The punctual elements (sources/concentration points and users) are connected in various forms: concentration point-users (e.g., drinking water and electricity), users-concentration point (e.g., wastewaters), user-user (e.g., telecommunication and individual trips), and concentration-concentration points (e.g., public transport). This diversity of punctual elements and specific and concrete relationships among them shape the urban technical networks which can be characterized at three levels:
Material (constituent components-punctual elements and connections);Structural (configurations for associated graphs-topology and geometry);Functional (characteristic of transferred flows).


Regardless of the nature of the transferred flows, the way of connecting the punctual elements or the level of examination, all urban infrastructure networks have the same goal, namely to ensure the solidarity, synchronization, and organization of the urban space. This paper aims at the diachronic analysis of urban technical networks and to highlight the correlation between network resilience and the consequences on the urban environment.

In the first part of the paper, we defined the operational structures for studying the evolution of urban technical networks and the necessity of analyzing the relations between them and the urban system is argued. The general experience shows that there are connections between the evolution of urban technical networks and that of the city [1,4,6]. To distinguish these connections, one has to go over the individual approach in which the different urban technical networks are examined, based on the nature of the flows. All networks of urban technical infrastructure need to be systematically studied, and united by reference to the functions they provide for urban space.

Urban technical infrastructures are critical infrastructures that, after the 2000s, have been the subject of particular attention in developed countries [7,8]. Although there is no common definition of critical infrastructures, it is widely accepted that membership in this category is determined by the consequences for security, lifestyle, and public health that the temporary loss of the functions of these infrastructures (communications, electricity generation and distribution, gas and fuel storage and transport, water supply and distribution, passenger transport, emergency services, etc.) would produce to urban life [9,10].

In the second part of the paper, the risks and consequences for the functions of the networks and the territorial system are analyzed, arguing the need to study the resilience of urban traffic networks. Resilience in contrast to classical risk management, which traditionally refers to rules and programs for each possible cause of a malfunction, brings to the foreground the characteristics of a system that is capable of complying with operational requirements even when changes of a certain magnitude are identified in the environment [10,11,12]. There is a need to replace the traditional vision of linear causes through a more complex, integrative vision (multiple, combined, but not clearly identified causes that generate possible events) [13,14].

For urban traffic networks, the paper examines the correlations among network resilience, the magnitude of degradations (material, structural, and functional), and the consequences for users. Two key aspects of the traffic network behavior at risk are highlighted: (i) The ability of the network to transfer flows even when some of the functions (features) have been degraded, and (ii) the network’s capacity to recover the degradation caused by the disturbances and to return to normal operation, by possibly restoring the components that have been degraded.

Finally, an example is presented for characterizing the resilience of an urban transport network to negative phenomena that affect its performance and the dynamism of its states. 

The case study identifies the required probabilities of the network states considering additional information about the travel time. The principle of maximum entropy is effective in computing these probabilities, based on the travel time constraint provided by the users of the physical network or by the public transport operator.

## 2. The Urban System and Its Technical Networks: A Systemic Examination

The dynamics of the city itself must be synchronous to the dynamics of networks, even if their limits could not overlap. The urban technical networks provide inner and outer exchanges of the city. Most often, urban dynamics research is aimed towards one of the subsystems or only one of the subsystem specific aspects. However, interactions among subsystems require systemic examination of urban dynamics.

*Urban Dynamics* [15], exploring the long-term consequences of urban policy decisions in great cities, qualifies the urban system as a robust one, less sensitive to changes in key parameters or inputs, for which the response to urban policies is slow and evolutionary. Nowadays, the city is assimilated to a self-regulating system that strives to achieve a balance for each of its subsystems, no matter how they evolved over time [16].

Relative to the temporal scale, urban subsystems evolve differently. Technical infrastructures and land use are subjects of very slow changes, areas for residences and activities are slowly evolving, and occupations, housing needs, and spatial mobility are more dynamic while people and the movement of goods are quite instantaneous [17,18,19].

Urban technical infrastructures (infrastructure for civil and industrial construction and traffic/circulation infrastructure), in addition to a long-life and very slow changes over time, require considerable investments, suffer from the inertia of the invested capital, indivisibility, and the gradual assimilation of the new capacities. These involve the intervention of public powers and the need for the socio-economic substantiation of the intervention priorities subject to budgetary constraints.

The ensemble of the urban technical network can be represented by an operational structure ℛ [5]:
(1)ℛ=[〈R〉|〈F〉]
where 〈R〉 is the aggregate of the sets of urban technical networks; 〈F〉 is the aggregate of the operators of the structure ℛ, that is, the set of transfer flows inside the network.

The aggregate of the sets of urban technical network contains the set $\left\{R_{m}\right\}$ that transfers material flows, the set $\left\{R_{e}\right\}$ transferring energy flows, and the set $\left\{R_{i}\right\}$ transferring information flows. 

For each of the sets of the aggregate 〈R〉, the overlapping of networks of the same nature can identify properties (connexity, connectivity, homogeneity, isotropy, and nodality) that measure the performances in servicing the city. 

The aggregate 〈F〉, containing the set of material $\left\{F_{m}\right\}$, energy $\left\{F_{e}\right\}$, and information flows $\left\{F_{i}\right\}$ is the synthetic expression of the transfer among the elements of the urban system, defined as S, by the operational form [5]:
(2)S=[〈S〉|〈A〉]
where 〈S〉 is the aggregate of sets of the city socio-economic systems and 〈A〉 the aggregate of sets of activities.

The aggregate of the sets of activities 〈A〉 determines the specific need for the transfer, the flows between the elements of 〈S〉, thus the “ex-ante” demand [20]. This need for transfer in conjunction with the aggregate of the sets of networks 〈R〉 determines the aggregate of the sets 〈F〉 of the transfer flows, thus the “ex-post” demand [20] or that part of the potential transfer need, which, under the existing constraints, could be satisfied.

The evolution of urban technical networks can be traced by studying the time dependency of S, especially that of 〈A〉 changes and their consequences in quantitative and qualitative changes of 〈F〉 with direct reflection in 〈R〉.

If the changes in the urban socio-economic environment, expressed through the operational structure S, cause only limited quantitative changes in 〈F〉, then the synchronous analysis can predict the evolution of ℛ which improves its performance by strategic and tactical actions or operative management.

If the quantitative changes of 〈F〉 exceed the saturation limits of the existing networks or their efficient exploitation, in any of the identified technologies, or if structural changes of 〈F〉 occur, then the prediction on ℛ must be the result of diachronic analysis. Extensions or reductions in each set $\left\{R_{m}\right\}$, $\left\{R_{e}\right\}$, $\left\{R_{i}\right\}$ are possible, and also the occurrence of substitution networks within one of the sets (e.g., from the tramway to the subway network) or even between the sets of the networks (e.g., from $\left\{R_{m}\right\}$ to $\left\{R_{e}\right\}$, by replacing the fuel transport for different urban socio-economic activities by its combustion in thermal power plants followed by the transport and use of electricity or from $\left\{R_{m}\right\}$ to $\left\{R_{i}\right\}$ by diminishing postal mail as a result of advances in information and communication technology).

It follows that, in particular, in a diachronic analysis imposed by scale dependencies of urban development that unconditionally includes the evolution of urban technical networks, the networks $\left\{R_{m}\right\}$, $\left\{R_{e}\right\}$, $\left\{R_{i}\right\}$ must be examined from a global perspective. The most evolved of each of the aggregate of sets with respect to the properties would give ℛ those properties that would characterize the autonomy, permanence, coherence, and organization of S [5].

The complexity of the city’s functions cannot be reduced to the study of urban technical networks treated holistically.

Reporting on the “ekistics” defined by Doxiadis [21] provides a more appropriate understanding of the complexity of the research required to correlate the aspects of urban life through the complementary contribution of various fields and disciplines (urban planning, architecture, economics, sociology, psychology, anthropology, culture, politics, etc.). Synthetically represented by the connections between the Doxiadis pentagon’s vertices (human, nature, society, networks, and constructions), the complexity of city life research is overwhelming [22] and explains the present and past diversity of the world’s cities.

Disturbing factors inside the urban system and the natural and anthropic environment affect its functioning, as opposed to the achievement of the proposed goals.

The entropic state of the city is specific to any system with self-regulation, which is the effect of the simultaneous action of two components: one that is oriented towards the fulfillment of the proposed goal (the positive component) and another one, the negative component, which tends to hinder or compromise the achievement of the proposed objectives. Endogenous and exogenous events classified as risks are the ones that negatively affect and disturb the functioning of the urban system.

## 3. Risks in Network Operation

Exogenous and endogenous random events adversely affect the operation of technical networks. The probability of occurrence of these events, interpreted as risks in preserving the structural and functional properties of the networks in the case of natural events (earthquakes, floods, storms, or extreme temperatures), does not depend on how the network was designed, realized, and managed. The magnitude of the consequences, i.e., the magnitude of the malfunctions produced by random events to networks, is interpreted as the individual and collective responsibility of those involved in the design, construction, and administration.

Of course, another is the situation of endogenous negative events of anthropogenic nature, in which both the probability of the occurrence of these events and their consequences are attributed to a wide diversity of technical, economic, social, political, and organizational actors [23,24,25,26]. Without claiming to be exhaustive, the enumeration of negative endogenous events of anthropogenic nature justifies the assertion. For example, can be identified [11,23]:
Economic risks, due to the fact that the technical networks are components of a competitive system under strong strategic (financial and economic) pressures that require important investments;Social risks, due to the fact that the technical networks present space-time usages, depending on the demand of the beneficiaries and the fact that the employees’ strikes can temporarily affect the functioning and therefore the performance of the networks;Technical risks, with origins in the dependencies among different types of networks and the technical degradation of equipment in time, or in disasters due to technological causes;Political risks caused by the diversion of flows as a result of conflicts between countries;Human risks associated with malevolent actions or terrorism;Organizational risks caused by malfunctions due to a lack of information, professional deficiencies, delays in decision making, etc.

Apart from the fact that the enumeration of risks of this nature cannot be interpreted as complete, we should also note that the placement of these anthropogenic risks in the category of internal ones (endogenous) must be admitted to be relative because it depends on how the network is interpreted: as a distinct part or as an indissoluble component of the system of technical networks of society, which in turn can be figured as a technical, socio-technical or socio-economic system (situations where the anthropogenic risks are either endogenous or exogenous).

Depending on the magnitude of the disturbing events, occurrence probability, and the extent of the malfunctions caused to the technical networks, different vulnerability levels, respectively aggravating factors of risk exposure are discriminative, such as spatial localization and network density, underground, ground or above ground localization, interdependence between networks, and the degree of network management concentration. The magnitude of the consequences is expressed by the severity of the damages, the duration of the malfunctions (operating interruptions), the cost of repairs or re-shifting, the consequences on the operation of other networks, the location of the disasters, the nature of the activities affected, the severity of the disorder caused, the society’s reaction to the recovering, etc.

The impact of the loss of technical network functionality is recorded at the territorial level through supply disruptions, degradation of living conditions, pollution, accidents, overloading, congestion, and degradation of civil constructions, i.e., through technical, economic, social, and environmental impacts (see Figure 1).

## 4. Engineering Resilience

### 4.1. Resilience and Vulnerability

Encountered in many engineering, ecology or psychology papers, the concept of resilience is multidisciplinary and polysemantic (ecological resilience, engineering resilience, socio-ecological resilience, and other disciplinary resilience, along with systemic resilience), with multiple and sometimes contradictory meanings [10,11,12]. Engineering resilience is defined by the ability of a system to maintain a steady-state of stability and having the strength and speed of return to the equilibrium position [12], while ecological resilience recognizes that the system maintains its functions and structure through various states of stable and unstable equilibrium [27,28,29]. In economic sciences, resilience has different meanings, from full system preservation to structural and qualitative renewal [12,28,30].

With all the possible sources of ambiguity, resilience is almost accepted as a new paradigm. Being resilient is supposed to be: robust, flexible and adaptable, redundant, diversified and effective, autonomous in a collaborative environment, able to learn from the past, and to cope with future uncertainties.

The requirements are listed from Holling’s first work in the field of ecology [31], resulting in a disciplinary resilience that has extended over time to systemic, socio-economic, social, and environmental resilience. But, polysemy associated with the multidisciplinary use of the concept persists [32].

The use of the concept of “engineering resilience” in the field of operational safety is also accompanied by a logical change in approaches [11]. In the field of risk management, resilience appears as a new paradigm, a paradigm of complexity, suitable to the study of complex events using particular components of adaptation, learning, persistence, stability, resistance, etc., which resilience puts in relationships.

The concept of resilience used in risk management coexists in parallel with vulnerability (fragility). Relationships between them derive directly from the definitions [13]. It appears that they are in a strongly disputed opposition that could, in fact, be summed up via two interpretations. The first one states that resilience is a positive attribute of the system that needs to be strengthened and enhanced, vulnerability is a negative attribute whose reduction should be considered. The second states that resilience and vulnerability are in a reciprocal relationship, like two sides of the same coin.

Both interpretations are criticized and disputed because resilience can also express negative aspects, as it should not be thought of as being necessary to reduce vulnerability to enhance resilience and vice versa.

Without continuing with the reference to the dispute between vulnerability and resilience in various fields (more significant in the social field), we conclude by saying that the two concepts are not necessarily opposed, that none of the concepts encompasses the other and both apply equally well in technical and social systems. The relative usefulness of each in the study of systems as well as clear boundaries between them are difficult to establish. However, it is proven that the use of the concept of systemic resilience (with physical and social dimensions) in the practical and theoretical risk studies is valuable and that in such studies resilience encompasses vulnerability. More specifically, resilience and vulnerability have outlined the concept of resilient vulnerability that brings together the negative connotations of vulnerability with the positive resilience to characterize a dynamic and reactive process of a disruptive system.

Resilient vulnerability suggests not only the possible degradation of system components, but also how they adapt to disturbances (proactive, reactive, and post-active) [33].

In the science of risk prevention, resilience can be defined as the ability of a system to absorb disturbances and recover the functions that have been affected by these disturbances. By definition, two terms are relevant: absorption and recovery, which remain rather vague and are therefore sources of ambiguity. For example, for an urban system, it is necessary to define the meaning of the absorption of a disturbance (which in turn must be specified) as well as the significance of the recovery; both charged with ambiguity.

The analytical study of the risks associated with the functioning of the urban system and in particular the systemic study, with reference to endogenous or exogenous random events (disturbances), to material damages and malfunctioning and to the repairing of the components and the whole urban system, must lead to the meaning of “absorption” and “recovery” at different time scales. In this approach, the complementarity between vulnerability and resilience becomes relevant.

### 4.2. Resilience of Urban Traffic Networks

The magnitude of degradations registered by the technical networks of infrastructures due to exogenous and endogenous hazards is gradually revealed at a material, structural, and functional level.

Correlations between the magnitude of degradation and network resistance determine the consequences of risks from a material to a structural and functional level. The properties of the physical network and of the public transport service network are defined for the level of propagation of negative random events affecting the network (see Figure 2).

Other trafficking features (in addition to connexity) can be crucial for the extent of degradations and for the consequences experienced by the beneficiaries (reduction of speeds, comfort, and convenience, an increase of traffic-related risk, damage to homogeneity and isotropy or even to the nodality of the network with results in modifying itineraries).

Figure 2 illustrates two essential aspects of the risk-carrying behavior of the traffic network. The first one refers to the ability of the network to play its role in transferring flows even when some functions (features) have been degraded. The network through resilience and absorption capabilities responds to disturbances while continuing its mission in the territorial system it serves. The second one concerns the network’s ability to recover the damages caused by disturbances and to return to normal (initial or improved) operation by restoring the components that have been degraded.

The two aspects are confirmations of the various definitions of resilience that keep resistance, absorption, and recovery as major elements:
The feature of a system that allows it to compensate for disturbances, making it possible to continue operating until the infrastructure is seriously damaged or destroyed [34];The ability of the system to keep its service level (normal) or to return to this level within a limited time frame [35];The ability of the system to adapt to variable and unexpected operating conditions, but without catastrophic destruction [36];The ability of the system to absorb the consequences of disturbances, reduce their impact, and maintain flow transfer [37,38];The ability of a network to maintain or restore acceptable operation despite disturbances [39].


The risk behavior defined by a scenario, with a specific probability and a certain intensity of the consequences, depends on a number of network resilience factors (redundancy, diversity, effectiveness, component autonomy, strength, collaboration, adaptability, mobility, security, and recovery capacity) for which quantitative indicators should be established. Because of the large number of these indicators, even aggregation and normalization are difficult.

For traffic networks, resilience depends both on availability and accessibility (i.e., internal resilience of the network) as well as user perceptions and the cost of travel/transport (i.e., user resilience). Sometimes these factors can quickly turn into a concept that is inoperable for various reasons. As a result, analytical treatment may contradict the systemic nature of the concept of resilience.

In front of this complexity of quantification of resilience, a particular focus on some or a single factor of resilience is used. Most often, the redundancy factor. Difficulties are not entirely removed because redundancy also depends on many other factors. Hence, heuristic algorithms are needed to obtain redundancy with fewer constraints.

For traffic networks, resilience is opposed to traffic congestion, for which computational processes that address the speed of traffic (compared to the free flow) and user delays are developed.

Under formal, mathematical terms, resilience is interpreted as a property of the graph associated with the network. From this perspective, the resilience of the graph corresponds to the minimum number of components (arcs or nodes) that either suppressed (or added) or can alter a property of the graph. Resilience is an indicator of the graph associated with the network, but the study of network resilience cannot be limited to what could be obtained from the graph study. The resilience, absorption, and recovery capabilities of the traffic network are the components whose investigation should result in network resilience indicators.

## 5. Network States Dynamics

The loss of functionality of some technical network elements, as well as the connection between two nodes of the traffic network, is interpreted as being temporary. The vital role of the technical networks for the transfer of flows to the benefit of the territorial system activities justifies the most promising interventions to remove as soon as possible the structural and functional degradations registered by the network.

The network’s resilience to the negative events affecting its performance and the interventions (preventive, corrective, and repair maintenance) to restore the network’s functions reveals a dynamic on a time scale appropriate to the network states. 

In order to study the dynamics of a traffic network, three states of the network are selected:
RI—the initial network;RC—the network that maintains its connexity;RN—the non-connex network (as a result of the severe degradation suffered).


If we are able to estimate at that time scale the probability of the network to get from one state to the next, then we have the possibility to estimate the probability that after a number of periods the network will be in one of the states.

To measure the periods/stages in which the network gets from one state to another, one defines a generating function of a time sequence $\left\{a_{t}\right\}$ using the Z-transform:
(3)$$A\left(z\right)=a_{0}+a_{1}z+a_{2}z^{2}+\cdots =\sum_{t=0}^{\infty }a_{t}z^{t},$$
where the coefficients $a_{t}$ represent the probability of the network to be in a specific state.

Transition processes from one state to another may be associated with a signal-graph with estimated probabilities of preserving the state and switching to another state (see Figure 3). The links of the graph in Figure 3 are associated with the Z-transform of the probability (the values used in the graph are increased relative to the real ones, to facilitate the interpretation of the computing results).

Let’s assume that incidentally, at the beginning of the analysis, the network is in the RC state and we compute the probabilities that after a certain number of periods the network should be in the same state.

To compute these probabilities, we introduce a dummy node, $y_{0}$, linked to the RC node (the observed network state) through a fictional link with transmittance value 1.

The transmittance $T_{y_{0}-\mathrm{RC}}$ could be written as a generating function of a time sequence $\left\{a_{t}\right\}$ (see Equation (3)), where the coefficients $a_{t}$ represent the probabilities of the network to be in the state RC after t time periods (t = 0, 1, 2, …).

In order to reduce the signal-graph in Figure 3, Mason’s gain rule [40] is suitable for computing the transfer function from a source node (S) to a final node (T):
(4)$$T_{S-T}=\sum_{j}\Pi_{j}\frac{\Delta_{j}}{D},$$
where $\prod_{j}$ is the path transmittance of the j-th forward path between the nodes S and T. D is the determinant of the graph:
(5)$$D=1-\sum L_{i}+\sum L_{i}L_{k}-\sum L_{i}L_{k}L_{l}+\ldots ,$$
where $L_{i}$ is the loop transmittance of each closed-loop, $L_{i}L_{k}$ is the product of the loop transmittances of any two non-touching loops (no common nodes), $L_{i}L_{k}L_{l}$ is the product of the loop transmittances of any three pairwise non-touching loops, etc. $\Delta_{j}$ is the cofactor value of D for the j-th forward path, with the loops touching the j-th forward path removed.

According to the graph in Figure 3:
(6)$$D=1-1.7z+0.8\;z_{}^{2}-0.1\;z_{}^{3}.$$


There is only one direct path between $y_{0}$ and RC, namely:
(7)$$\Pi_{1}=1\;\mathrm{and}\;\Delta_{1}=1-\left(0.8\;z+0.3\;z+0.02\;z^{2}\right)+0.24\;z^{2}=1-1.1\;z+0.22\;z^{2},$$
thus:
(8)$$T_{y_{0}-\mathrm{RC}}=\frac{1-1.1\;z+0.22\;z^{2}}{1-1.7\;z+0.8\;z_{}^{2}-0.1\;z_{}^{3}},$$
which can be written as a generating function of time sequence:
(9)$$T_{y_{0}-\mathrm{RC}}=1+0.6\;z+0.44\;z_{}^{2}+0.368\;z_{}^{3}+0.333\;z_{}^{4}+0.316\;z_{}^{5}+0.308\;z_{}^{6}+0.304\;z_{}^{7}+\cdots ,$$


As could be observed, after 6-7 time sequences, the probability of the system to be in the RC state stabilizes around the value P(RC)=0.30.

Of course, it is interesting to estimate the probability for the network to maintain its connexity, which is equivalent to finding the coefficients $z^{t}$ in $T_{y_{0}-\mathrm{RC}}+T_{y_{0}-\mathrm{RI}}$, where $T_{y_{0}-\mathrm{RI}}=\frac{0.3\;z-0.07\;z^{2}}{D}$ is similarly computed as $T_{y_{0}-\mathrm{RC}}$.

It results in:
(10)$$T_{y_{0}-\mathrm{RC}}+T_{y_{0}-\mathrm{RI}}=\frac{1-0.8\;z+0.15\;z_{}^{2}}{1-1.7\;z+0.8\;z_{}^{2}-0.1\;z_{}^{3}},$$
respectively:
(11)$$T_{y_{0}-\mathrm{RC}}+T_{Y_{0}-\mathrm{RI}}=1+0.9\;z+0.88\;z_{}^{2}+0.876\;z_{}^{3}+0.875\;z_{}^{4}+0.875\;z_{}^{5}+0.875\;z_{}^{6}\ldots ,$$


The system preserves its connexity with a probability of P(RI)+P(RC)=0.875.

Obviously, $T_{y_{0}-\mathrm{RC}}+T_{y_{0}-\mathrm{RI}}+T_{y_{0}-\mathrm{RN}}$ corresponds to a generating function where all coefficients of $z^{t}$ are 1 (any of the three states is possible).

Since:
(12)$$T_{y_{0}-\mathrm{RC}}+T_{y_{0}-\mathrm{RI}}+T_{y_{0}-\mathrm{RN}}=\frac{1-0.7\;z+0.1\;z_{}^{2}}{1-1.7\;z+0.8z_{}^{2}-0.1\;z_{}^{3}},$$
results in:
(13)$$T_{y_{0}-\mathrm{RC}}+T_{y_{0}-\mathrm{RI}}+T_{y_{0}-\mathrm{RN}}=1+z+z^{2}+z^{3}+\cdots ,$$
defining the complete probability space.

The uncertainty is expressed quantitatively by the information which we do not have about the state of the urban transport network, thus the entropy of the system:
(14)$$H=P\left(\mathrm{RC}\right)\log_{2}\frac{1}{P\left(\mathrm{RC}\right)}+P\left(\mathrm{RI}\right)\log_{2}\frac{1}{P\left(\mathrm{RI}\right)}+P\left(\mathrm{RN}\right)\log_{2}\frac{1}{P\left(\mathrm{RN}\right)},$$
where P(X) is the probability of the network being in state X.

Based on the evolution in time of the probabilities of the three states of the network, the variation of the entropy H is depicted in Figure 4. After seven time periods, the network entropy is getting stabilized at the value H=1.355 [bit], and the corresponding probabilities of the network states are: P(RI)=0.575, P(RC)=0.3, and P(RN)=0.125.

If one of the probabilities is equal to 1, then all the other probabilities are 0 and the entropy evaluates to 0 bits. Due to the possible risks outlined in paragraph three, it is impossible to maintain indefinitely the system in the desirable states RI or RC that provide the network connexity. On the other hand, P(RN)=1 means that the network has lost its resilience and therefore the resistance, absorption, and recovery are no longer effective.

It is a property of the entropy that it has its maximum value when all probabilities are equal. If there is no additional information about the network states, the maximum entropy is $H_{\max}=\log_{2}3=1.585\;\left[\mathrm{bit}\right]$. 

The uncertainty of different systems can be compared using the relative entropy $h_{r}=\frac{H}{H_{\max}}$. Smaller values for the relative entropy are desirable, but for the specific urban traffic network, this is feasible when P(RI)+P(RC)≫P(RN).

## 6. Discussion

Users are not able to directly appreciate the resilience of the transport network through the states transition probabilities, but through the performance of the network and/or the transport service. As Figure 2 shows, users may experience different consequences according to the robustness, absorption, and recovery capabilities of the traffic network. Individual consequences should be aggregated by the network manager in order to adopt appropriate measures for maintaining the network in a fair balanced state.

As an example, Figure 5 illustrates the three states (RI, RC, and RN) for the Bucharest subway network. A dysfunctional section (Figure 5b) still allows the network connexity, but some users have to travel through bypass routes. A dysfunctional station that is a network node (Figure 5c) is translated into the loss of the functionality for a part of the network, and some users should address partially or totally to the services provided by other networks. The users identify the three states of the network by the way they are reflected in the individual travel time. The changes in the average travel time at the network level can be used by the manager in defining the appropriate probabilities for each of the network states.

We assume the following values of the travel time for the three states of the network: ${\bar{t}}_{\mathrm{RI}}=25\;\min$, ${\bar{t}}_{\mathrm{RC}}=28\;\min$, and ${\bar{t}}_{\mathrm{RN}}=40\;\min$. If the network loses its connexity (RN), as it is stated in paragraph two, the flows are transferred using alternative technical networks, but the travel time increases. 

The average travel time in the network $\left(\bar{t}\right)$ is stated:
(15)$$\bar{t}=P\left(\mathrm{RC}\right){\bar{t}}_{\mathrm{RC}}+P\left(\mathrm{RI}\right){\bar{t}}_{\mathrm{RI}}+P\left(\mathrm{RN}\right){\bar{t}}_{\mathrm{RN}}.$$


The principle of maximum entropy [41] can be used to find the probability distribution which leads to the highest accepted value for the uncertainty considering users expectancies. The principle of maximum entropy states that the probability distribution which leaves the largest remaining uncertainty (i.e., the maximum entropy) consistent with the constraints is the solution of the problem:
(16)$$\max H=P\left(\mathrm{RC}\right)\log_{2}\frac{1}{P\left(\mathrm{RC}\right)}+P\left(\mathrm{RI}\right)\log_{2}\frac{1}{P\left(\mathrm{RI}\right)}+P\left(\mathrm{RN}\right)\log_{2}\frac{1}{P\left(\mathrm{RN}\right)},$$
subject to:
(17)$$\begin{array}{c} 0=\bar{t}-\left(P\left(\mathrm{RC}\right){\bar{t}}_{\mathrm{RC}}+P\left(\mathrm{RI}\right){\bar{t}}_{\mathrm{RI}}+P\left(\mathrm{RN}\right){\bar{t}}_{\mathrm{RN}}\right),\;\mathrm{and}\\ 0=1-\left(P\left(\mathrm{RC}\right)+P\left(\mathrm{RI}\right)+P\left(\mathrm{RN}\right)\right). \end{array}$$


The technique of Lagrange multipliers is used to find the probabilities that provide the maximum acceptable uncertainty, allowing the network to operate according to its functionalities.

Considering additional information about the required average travel time in the network $\left(\bar{t}\right)$, the maximum uncertainty and the distribution of the network state probabilities for two scenarios are given in Table 1. 

The first scenario is a more relaxed one, having an average travel time constraint equal to 29 min. This is achieved if the probability of network connexity is P(RI)+P(RC)=0.808. The network entropy is $H_{\text{max}1}=1.510$ [bit], that means a higher uncertainty compared to the current network.

In the second scenario, a harder condition is imposed on the average travel time $\left(\bar{t}=26\;\min\right)$. According to the new constraint, the connexity of the network should be characterized by a probability P(RI)+P(RC)=0.992, and a maximum uncertainty $H_{\text{max}2}=0.936$, a smaller value compared to the current network. To accomplish the second scenario, the risks that bring the network in the state of non-connexity have to be reduced, and/or the recovery time and implicitly the probability to remain in the state of non-connexity have to decrease.

## 7. Conclusions

Urban technical infrastructures, regardless of the nature of the flows (materials, energy, and information), support the quality of life of the inhabitants. Therefore, both in the design and the performance analysis, the infrastructures should be treated systemically. The complementary contribution of the various fields and disciplines (urban planning, architecture, economy, sociology, psychology, anthropology, culture, politics, etc.) is necessary to explain the past and present diversity of urban forms, dimensions, and structures.

The functionality of the urban technical infrastructures is influenced by endogenous and exogenous disturbance factors, qualified as risks. The entropic state of the city, specific to any system with self-regulation, is characterized by the effects of the simultaneous action of two adverse components—one positive, oriented towards the fulfillment of the proposed goal and the other negative, which tends to compromise the completion of the objectives. This makes almost impossible any attempt to estimate the separate and/or conjugate effects of the different risk categories (economic, social, technical, political, human, and organizational) on the performance of any of the urban technical infrastructures. The difficulty is also accentuated by the inter-correlation among the vulnerabilities of the various networks. 

Therefore, in the study of the risk behavior of urban traffic networks, intensely publicized components of the whole urban technical networks, especially through the negative external effects of motorized traffic, we preferred a complex, integrating vision (with multiple, unspecified, and wide-ranging combined causalities that generate identifiable effects) that involves the concepts of resilience, engineering resilience, and resilience vulnerability, as reflected in the recent literature.

The consequences of the degradation caused to the urban technical networks by the risk factors, and to the urban traffic networks, in particular, are recorded at the material, structural, and functional level. They are reflected by changes in the properties of the physical network (relevant for individual travel performances) and of the service network (relevant for urban public transport users).

In identifying these consequences from the multitude of existing factors, for the characterization of the resilience of the traffic networks, we focused on the resistance and absorption capacity, both conferred by the redundancy incorporated in the network through the design and implementation.

The urban transport network is characterized by three states: (i) RI state unaffected in terms of connectivity, (ii) RC state with network connectivity affected, but still connex, and (iii) RN state for the non-connex network. The dynamics of the network states have been studied using a generating function of time sequence and the signal graph associated with the network states, on which the Z-transform of the probabilities of transition are the links transmittance. It has been shown that, regardless of the initial state, after a certain number of periods the probabilities of each state are stabilized (the stationary regime is reached).

This case study identifies the required probabilities of the network states considering additional information about the travel time. The principle of maximum entropy is effective in computing these probabilities, based on the travel time constraint provided by the users of the physical network or by the public transport operator.

## Figures and Tables

**Figure 1 entropy-21-00886-f001:**
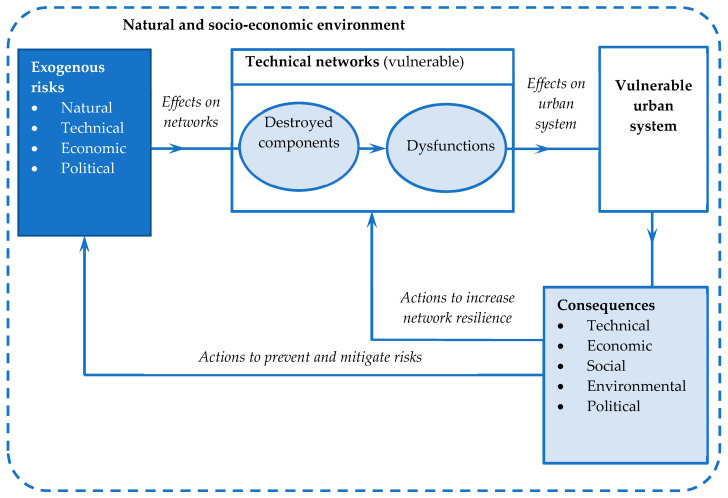
Risks and consequences for network and urban system.

**Figure 2 entropy-21-00886-f002:**
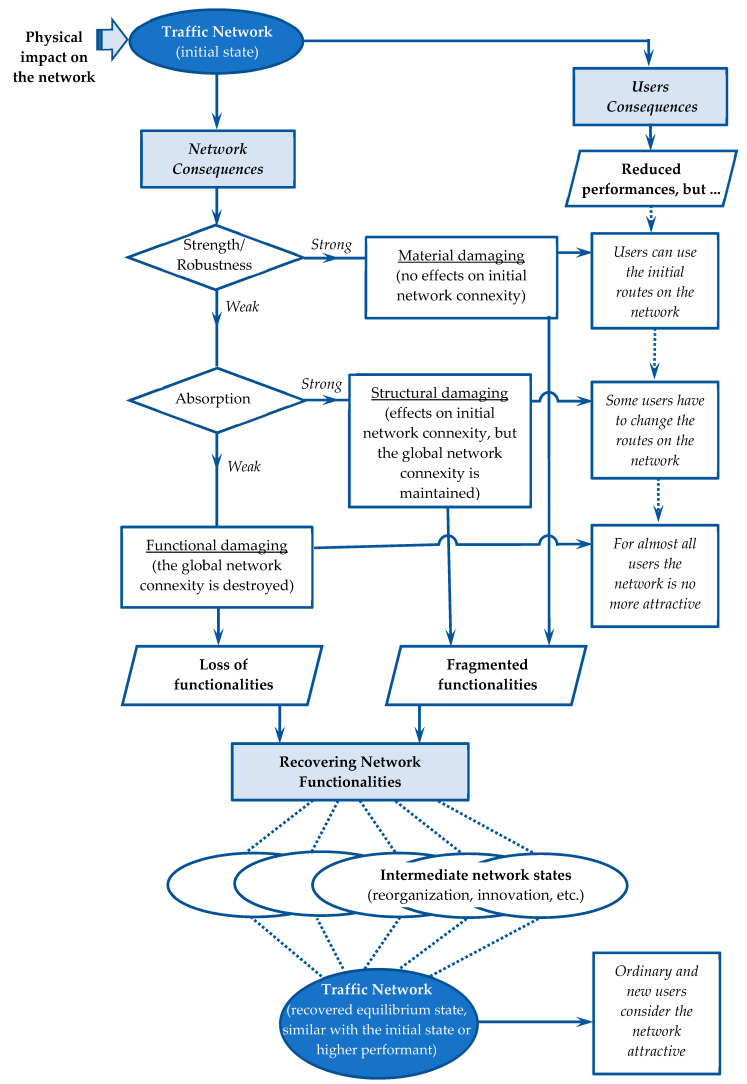
Correlations between network resilience, extent of degradation (material, structural, and functional), and consequences for users.

**Figure 3 entropy-21-00886-f003:**
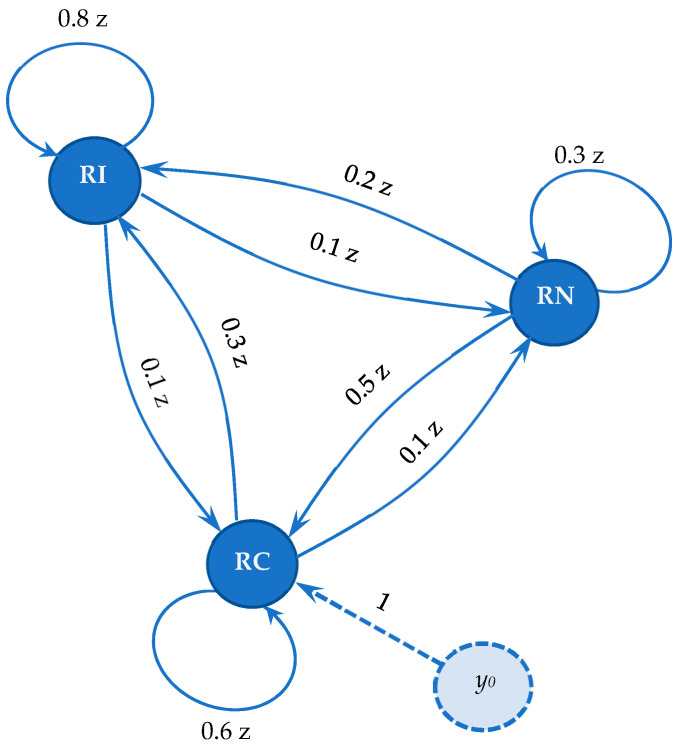
The signal-flow graph associated with the network transition states.

**Figure 4 entropy-21-00886-f004:**
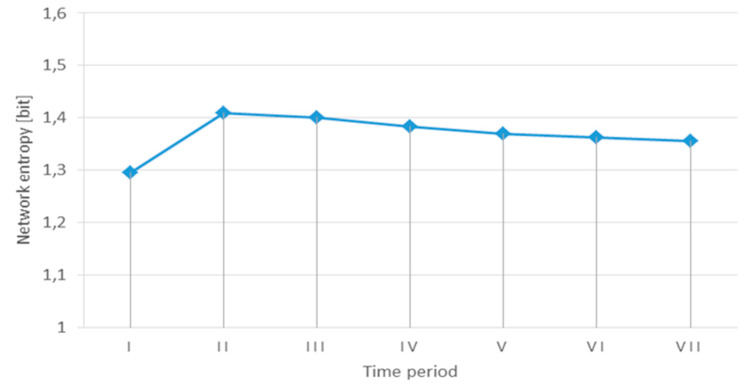
Evolution of the urban transport network entropy.

**Figure 5 entropy-21-00886-f005:**
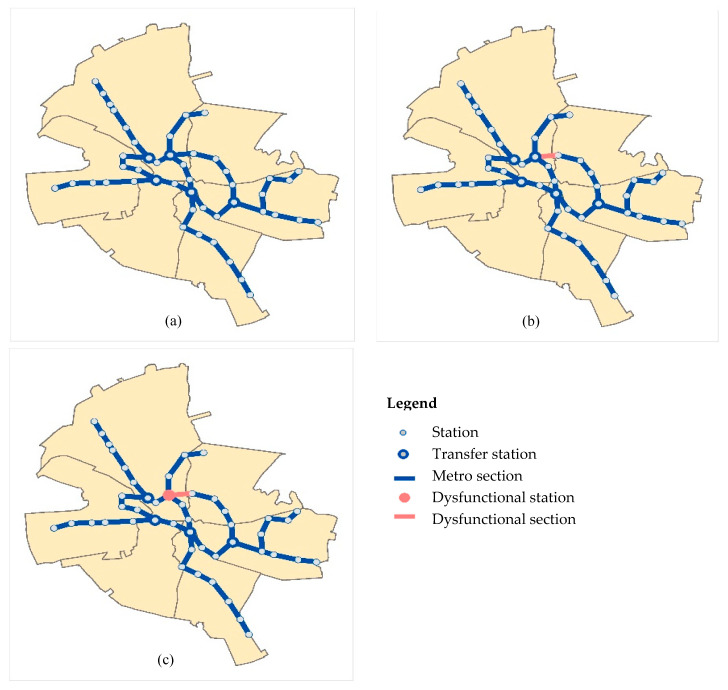
Examples of metro network states. (**a**) Initial fully functional network; (**b**) network with local dysfunctionalities, maintaining connexity; (**c**) network with local extended dysfunctionalities and isolated elements (non-connexity).

**Table 1 entropy-21-00886-t001:** Maximum acceptable uncertainty in the urban transport network.

	Scenario 1 $\bar{t}=29\;\min$	Scenario 2 $\bar{t}=26\;\min$
P(RI)	0.436	0.700
P(RC)	0.372	0.292
P(RN)	0.192	0.008
H [bit]	$H_{\text{max}1}=1.510$	$H_{\text{max}2}=0.936$

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
