# Peer review of "Resilience of Urban Technical Networks"

_entropy, 2019, doi:10.3390/e21090886_

Round 1

Reviewer 1 Report

This research deals with the risk vs resilience statistical and quantitative analysis of complex urban networks. Authors specifically investigates on traffic/transportation networks implementing a custom application of the Ekistics framework using Entropy function and probabilistic approaches linked to model/network states. The states are characterized by simplified assumptions of the two extreme fully working and disrupted states and one intermediate.

A case study of a metropolitan system of Bucharest is shown to depict such framework and an analytical approach is proposed to infer the probabilities and entropy quantifications for the pre-identified state variables.

The topic of this work is of interest for this journal and authors demonstrate to have solid and sound knowledge of the topic. The writing is fairly accurate and correct for non English mother tongue speaker, apart some minor issues, and the structure of the manuscript is also proper for the topic and editorial setting.

Nevertheless, I have some minor remarks and general comments I’d like authors to address before resubmitting this work for potential publication as well as some specific comments that are provided as comments in the attached commented-manuscript PDF.

General comments

There are some predefined assumptions, like optimal versus disrupted travel time, that impact the generability and replicability of the proposed model. Same remark applies for the system states. The three states, depicted in the theoretical model and its applications, can be argued and may also not applicable to other urban networks. Authors should explain how those pre-defined scenarios and parameters are selected and suggest to the reader how to replicate the model application on a different transport network, and also to a different network type (considering that several times authors claim this framework can be applied to other network types). In Section 2 in the description of “Exogenous and endogenous random events” I find that the description remains too theoretical and is not matching real case event authors are mentioning. See also the specific comments on factors affecting urban network exposed to flood risk. The network itself in this case, the streets, become surface river flow features, and modify the dynamics and impact of the event. I suggest authors to enrich this section by inserting some real case descriptions. There is a full section on resilience and on engineering resilience in particular, but I don’t find relevant the resilience evaluation in the proposed modelling framework. See also next comments on the disconnection between the introduction, the model and the results. I find that the introduction is quite rich and also lengthy in mentioning the concept of urban networks, risk and resilience, but I found that the modelling and results sections don’t properly implemented the introduced concepts. This makes the manuscript quite extended in the framework description as well as in the flow charts of figure 1 and figure 2, but then quite under-performing in the modelling and results sections.

General comments

See attached PDF with comments

Author Response

We would like to thank for your helpful feedback and the great efforts in revising our submission.

We have considered very carefully your suggestions and comments.

Point by point responses (in red) to the comments (in black) are listed in the attached file.

Thank you!

Reviewer 2 Report

I  suggest to include some results in the abstract.

Figure 1 is very interesting.

Methods: section 2 and 3 provide a framework to develop Figure 2, that is used in the method. The method and application are presented in section 5, in the same way. In my opinion, this way brings more understanding to the reader. Despite this method is conceptual (in my opinion), the result is a way to identify the resilience of the urban network in a simple way. This type of analyses could support urban transport planning since identifying the links without resilience, these links could be supported with new projects, mainly in developing countries. The discussion brings these insights from the results. I understood the discussion is, also, the conclusion. The authors could bring insights to readers apply the method proposed in practical situations.

My suggestions to authors bring connection between section 2, 3, 4 and 5. The section 2, 3 and 4 is necessary to understand the section 5. However, don't have a good connection. Maybe, change the title of the section could bring more understanding to the reader. For example, section 5 is called "Network States dynamics. Results" This section bring more than Results. Bring an application of Figure 2. Finally, the discussion section could call a conclusion. It is a conclusion of the paper.

Author Response

(The authors gave the same response as above.)

Reviewer 3 Report

General comments

The content of this paper is rather vague, and the method of mathematical modeling is inappropriate. First, polynomials are not suitable for modeling complex systems. Polynomials have no specific mathematical structure. Therefore, the parameters of a polynomial bear no system meaning. Secondly, there is no substantial relationship between this study and entropy. Although the authors used entropy as measure in several places of this article, it seems that the decorative significance of entropy is greater than the substantive significance. In addition, the writing of this article is sloppy.

Specific points

Mathematical expressions. The uses of mathematical symbols are in confusion to some extent, including roman type and italic type, superscript and subscript. Generally speaking, functions, numbers, parentheses, and so on, should be written in roman type, while variables and parameters should be expressed in italic type.

Model selection. Polynomials are suitable for interpolation rather than for system modeling. The reason is that polynomials have no specific mathematical structure. In theory, many nonlinear equations can be expanded into polynomials by Taylor series and other methods.

Abstract. The abstract had better been revised according to the following structural arrangement: background, methods, results or findings, conclusions or significance.

Discussion. The analytical process of discussion is not clear. The section of Discussion in a paper is generally involved with 3 or 4 parts: (1) main points, which response to the questions put in introduction; (2) comments on related studies or problems; (3) shortcomings or deficiency in study method or process; (4) conclusions, which can be separated to make the final section. [See: Robert A. Day, Barbara Gastel. How to Write and Publish a Scientific Paper (Sixth Edition). Cambridge University Press, 2003]

Conclusions. The conclusions should be made clearer in expressions and meanings. The important conclusions should be given three times in a paper: once in the Abstract, again in the Introduction, and again (in more detail probably) in the Discussion (if it contains a Conclusion paragraph) or Conclusions (if this part is separated as the final section).

Author Response

(The authors gave the same response as above.)

Round 2

Reviewer 1 Report

My general and specific comments have been addressed. To my view and knowledge the manuscript can be accepted as it is.

Reviewer 3 Report

The quality of this paper has been improved after revising by referring to reviewers’ comments. The authors tried their best to deal with the questions raised by the reviewers. Although there are still some methodological problems, maybe it can be published for further academic discussion.